# Muscle-specific economy of force generation and efficiency of work production during human running

Sebastian Bohm[1,2]*, Falk Mersmann[1,2], Alessandro Santuz[1,2], Arno Schroll[1,2], Adamantios Arampatzis[1,2]

[1]Humboldt-Universität zu Berlin, Department of Training and Movement Sciences, Berlin, Germany; [2]Berlin School of Movement Science, Humboldt-Universität zu Berlin, Berlin, Germany

**Abstract** Human running features a spring-like interaction of body and ground, enabled by elastic tendons that store mechanical energy and facilitate muscle operating conditions to minimize the metabolic cost. By experimentally assessing the operating conditions of two important muscles for running, the soleus and vastus lateralis, we investigated physiological mechanisms of muscle work production and muscle force generation. We found that the soleus continuously shortened throughout the stance phase, operating as work generator under conditions that are considered optimal for work production: high force-length potential and high enthalpy efficiency. The vastus lateralis promoted tendon energy storage and contracted nearly isometrically close to optimal length, resulting in a high force-length-velocity potential beneficial for economical force generation. The favorable operating conditions of both muscles were a result of an effective length and velocity-decoupling of fascicles and muscle-tendon unit, mostly due to tendon compliance and, in the soleus, marginally by fascicle rotation.

**\*For correspondence:**
sebastian.bohm@hu-berlin.de

**Competing interest:** The authors declare that no competing interests exist.

## Introduction

During locomotion, muscles generate force and perform work in order to support and accelerate the body, and the activity of the lower-limb muscles accounts for most of the metabolic energy cost needed to walk or run (*Kram and Taylor, 1990*; *Kram, 2000*; *Dickinson et al., 2000*). Running is characterized by a spring-like interaction of the body with the ground, indicating a significant conversion of the body's kinetic and potential energy to strain energy - via the elongation of elastic elements, mainly tendons - that can be recovered in the propulsive second half of the stance phase (*Dickinson et al., 2000*; *Roberts and Azizi, 2011*; *Cavagna et al., 1964*). In addition, the elasticity of tendons influences the operating conditions of the muscles, which in turn are associated with their metabolic cost (*Roberts, 2002*). For a given muscle force, the metabolic cost depends on the muscle's operating force-length and force-velocity potential (*Bohm et al., 2019*; *Bohm et al., 2018*; *Nikolaidou et al., 2017*) (fraction of maximum force according to the force-length [*Gordon et al., 1966*] and force-velocity [*Hill, 1938*] curves) because it determines the number of recruited muscle fibers and thus the active muscle volume (*Roberts, 2002*). This means that quasi isometric contractions close the optimum of the force-length curve, that is, with a high force-length-velocity potential, are theoretically most economical for generating a given force. During steady-state running, however, the human system does not perfectly conserve all the mechanical energy in each stride. Therefore, muscular work by active muscle shortening is needed to maintain the running movement, yet it increases the metabolic cost a) due to the reduced force-velocity potential, which will increase the active muscle volume for a given force (*Roberts and Azizi, 2011*), and b) due to the higher metabolic energy consumption

of each fiber when actively shortening (**Smith et al., 2005**; **He et al., 2000**). The active shorting range and velocity of a muscle during movements can be reduced by its tendon and, thus, an important benefit of tendon elasticity is a reduction in the metabolic cost of running.

At the muscle level, it has been shown that the triceps surae muscle group produces muscular work/energy during the stance phase of steady-state running (**Lai et al., 2014**). The soleus is the largest muscle in this group (**Albracht et al., 2008**) and does work by active shortening throughout the entire stance phase (**Bohm et al., 2019**; **Bohm et al., 2021**). In the first part of the stance phase, the performed muscular work is stored in the Achilles tendon as elastic strain energy. During the later propulsion phase, the tendon strain energy recoil contributes to the muscular energy production, suggesting an energy amplification behavior (**Roberts and Azizi, 2011**) within the triceps surae muscle-tendon unit (MTU) during running. On the contrary, the vastus lateralis muscle, as the largest muscle of the quadriceps femoris muscle group (**Mersmann et al., 2015**), operates nearly isometrically despite a lengthening-shortening behavior of the vastus lateralis MTU (**Bohm et al., 2018**; **Monte et al., 2020**). The almost isometric contraction suggests a negligible mechanical work production by the vastus lateralis during running and a spring-like energy exchange between body and vastus lateralis MTU, which promotes energy conservation (**Dickinson et al., 2000**; **Roberts and Azizi, 2011**).

The triceps surae and the quadriceps muscle group are considered to be crucial for running performance (**Arampatzis et al., 2006**; **Hamner and Delp, 2013**). The quadriceps femoris decelerates and supports the body early in stance while the triceps surae accounts for the propulsion later in the stance phase (**Dorn et al., 2012**; **Santuz et al., 2020**; **Hamner and Delp, 2013**). The soleus and vastus lateralis, as the largest muscles of both muscle groups, show marked differences in their morphological and architectural properties with shorter fascicles and higher pennation angles in the soleus (**Bohm et al., 2019**; **Maganaris et al., 1998**) compared to vastus lateralis (**Bohm et al., 2018**; **Marzilger et al., 2018**). Because of the long fascicles of the vastus lateralis, a unit of force generated by this muscle is metabolically more expensive (**Biewener and Roberts, 2000**) compared to the soleus. Our previous findings (**Bohm et al., 2018**) suggest that the vastus lateralis operates at a high force-length-velocity potential during running, which would indicate a fascicle contraction condition that could minimize the energetic cost of muscle force generation. The soleus muscle instead operates as a muscular work generator through active shortening, though close to the optimum of the force-length curve. Operating with increasing shortening velocity decreases the force-velocity potential according to the force-velocity relationship (**Bohm et al., 2019**; **Bohm et al., 2021**) and may increase the energetic cost of muscle force generation, marking a trade-off between mechanical work production and metabolic expenses. The enthalpy efficiency (**Barclay, 2015**) (or mechanical efficiency; **Hill, 1939**; **Hill, 1964**) quantifies the fraction of chemical energy from ATP hydrolysis that is converted into mechanical work and depends on the shortening velocity, with a steep increase at low shortening velocities up to a maximum at around 20% of the maximum shortening velocity ($V_{max}$) and a decrease thereafter (**Hill, 1939**; **Barclay et al., 1993**; **Hill, 1964**). Previous findings suggest that the soleus fascicles continuously shorten at a moderate velocity during the stance phase of running (**Bohm et al., 2019**), covering a range that corresponds to a high efficiency. Therefore, the soleus muscle may operate at fascicle conditions that would be beneficial for economical work/energy production.

The muscle fascicle behavior is strongly influenced by the decoupling of the fascicles from the MTU excursions due to tendon elasticity and fascicle rotation (**Azizi et al., 2008**; **Alexander, 1991**; **Zuurbier and Huijing, 1992**; **Wakeling et al., 2011**). The previously reported decoupling of the soleus muscle indicates that tendon elasticity and fascicle rotation affect the operating fascicle length and velocity during running (**Bohm et al., 2019**; **Werkhausen et al., 2019**); however, their integration in the regulation of the efficiency-fascicle velocity dependency is unclear. Regarding the vastus lateralis muscle, it was suggested that proximal muscles like the knee extensors feature less compliant tendons compared to the distal triceps surae muscles, thus limiting the decoupling between fascicles and MTU (**Farris and Sawicki, 2012**; **Biewener and Daley, 2007**; **Biewener, 2016**). However, in our previous study, we found significantly smaller vastus lateralis fascicle length changes compared to the vastus lateralis MTU (**Bohm et al., 2018**), indicating an important decoupling within the vastus lateralis MTU due to tendon elasticity.

The purpose of this study was to assess the soleus and the vastus lateralis fascicle behavior with regard to the operating force-length-velocity potential and enthalpy efficiency to investigate physiological mechanisms for muscle work production and muscle force generation during running. The

soleus muscle actively shortens during the stance phase at moderate velocities, which may match the plateau of the enthalpy efficiency-velocity curve, and operates close to the optimum of the force-length curve. Therefore, we hypothesized that the soleus muscle as a work generator operates at a high force-length potential and a high enthalpy efficiency, minimizing the metabolic cost of work production. On the other hand, the vastus lateralis muscle that promotes energy conservation seems to operate at a favorable length and almost isometrically. Thus, we hypothesized a high force-length and a high force-velocity potential that would reduce the metabolic energy cost of muscle force generation. In order to investigate the regulation of the efficiency and force potentials, we further quantified the length and velocity-decoupling of the fascicles from the MTU as well as the electromyographic (EMG) activity. Because of experimental constrains, the two muscles were measured in two groups, respectively.

## Results

There were no significant differences in the anthropometric characteristics between groups (age p=0.369, height p=0.536, body mass p=0.057). The experimentally assessed optimal fascicle length for force generation ($L_0$) of the soleus was on average 41.3 ± 5.2 mm and significantly shorter than $L_0$ of the vastus lateralis with 94.0 ± 11.6 mm (p<0.001). The forces that corresponded to $L_0$ of soleus and vastus lateralis ($F_{max}$) were 2887 ± 724 N and 4990 ± 914 N (p<0.001), respectively. Furthermore, the assessed $V_{max}$ was 279 ± 35 mm/s for the soleus, significantly lower than the $V_{max}$ of the vastus lateralis with 1082 ± 133 mm/s (p<0.001).

The stance and swing times during running were 304 ± 23 ms and 439 ± 26 ms for the soleus group and 290 ± 22 ms and 448 ± 30 ms for the vastus lateralis group (p=0.075, p=0.369). The EMG comparison showed that the soleus was active throughout the entire stance phase of running while the vastus lateralis was mainly active in the first part of the stance and with an earlier peak of activation (soleus 41 ± 5% of the stance phase, vastus lateralis 35 ± 4% of the stance phase, p<0.001, *Figure 1*). During the stance phase, the MTU of both muscles showed a lengthening-shortening behavior, but the vastus lateralis MTU started to shorten earlier (soleus 59 ± 2% of the stance phase, vastus lateralis 50 ± 2% of the stance phase, p<0.001, *Figure 1*). The soleus and the vastus lateralis fascicle length were clearly decoupled from the MTU length with smaller operating length ranges throughout the whole stance phase (*Figure 1*). The soleus fascicles operated at a length close to $L_0$ at touchdown and then shortened continuously until the foot lift-off (0.994–0.752 $L/L_0$, *Figure 1*). The operating length of the vastus lateralis fascicles remained closely above $L_0$ over the entire stance phase and was on average significantly longer compared to the soleus fascicles (soleus 0.899 ± 0.104 $L/L_0$, vastus lateralis 1.054 ± 0.082 $L/L_0$, p<0.001, *Figure 1*).

The stance phase-averaged force-length potential of both muscles was high and not significantly different (p=0.689, *Figure 2*). The average pennation angle of the soleus was significantly greater than that of the vastus lateralis (soleus 23.9 ± 5.1°, vastus lateralis 13.3 ± 1.8°, p<0.001) and increased continuously throughout stance, whereas it remained almost unchanged in the vastus lateralis (*Figure 1*). The average operating velocity of the soleus fascicles was significantly higher compared to the vastus lateralis (soleus 0.799 ± 0.260 $L_0$/s, vastus lateralis 0.084 ± 0.258 $L_0$/s, p<0.001), which showed an almost isometric contraction throughout stance. Consequently, the force-velocity potential (p<0.001) and thus the overall force-length-velocity potential (p<0.001) of the soleus was significantly lower compared to the vastus lateralis during the stance phase (*Figure 2*). However, the higher shortening velocity of the soleus was close to the optimum for maximum enthalpy efficiency, leading to a significantly higher enthalpy efficiency over the stance phase in comparison to the vastus lateralis (p<0.001, *Figure 3*).

The fascicle, muscle belly, and MTU length changes throughout stance as well as the resulting velocity decoupling coefficients (DC) are illustrated in *Figure 4* for both muscles, where $DC_{Tendon}$ quantifies the decoupling due to tendon compliance, $DC_{Belly}$ due to fascicle rotation, and $DC_{MTU}$ the overall decoupling of MTU and fascicles. There was a clear length and velocity-decoupling of MTU and belly due to tendon compliance in both muscles (*Figure 4*). The statistical parametric mapping (SPM) analysis revealed a significantly lower $DC_{Tendon}$ of the soleus compared to the vastus lateralis between 4% and 8% of the stance phase (p=0.032) since decoupling started later for the soleus. Between 20% and 57% of the stance phase (p<0.001) and between 65% of the stance phase until lift-off, the soleus $DC_{Tendon}$ was significantly higher than vastus lateralis (p<0.001, *Figure 4*). The $DC_{Tendon}$ averaged over

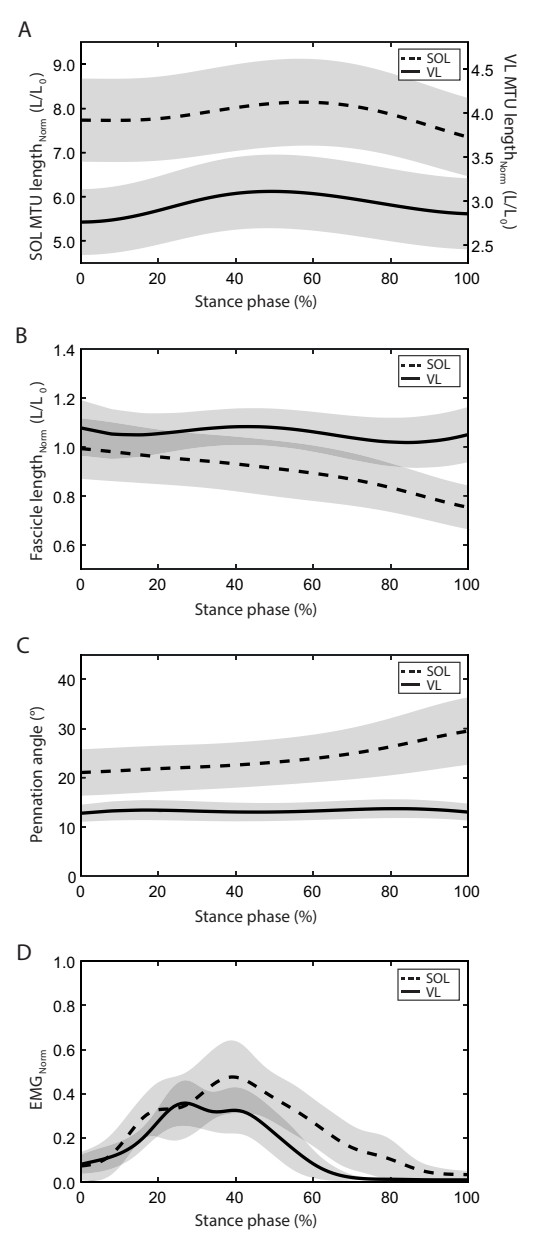

**Figure 1.** Soleus (SOL, n = 19) and vastus lateralis (VL, n = 14) muscle-tendon unit (MTU) length (**A**) and muscle fascicle length (normalized to optimal fascicle length $L_0$, **B**), pennation angle (**C**), and electromyographic (EMG) activity (normalized to a maximum voluntary isometric contraction, **D**) during the stance phase of running (mean ± SD).

The online version of this article includes the following source data for figure 1:

**Source data 1.** Numerical data represented in the graph 1.

the stance phase of the soleus was also significantly greater (p<0.001, *Table 1*). Furthermore, the velocity-decoupling of muscle belly and fascicles due to fascicle rotation progressively increased in the second part of the stance phase for the soleus but was negligible for the vastus lateralis (*Figure 4*). The soleus $DC_{Belly}$ was significantly higher from 33% of the stance phase until lift-off compared to the vastus lateralis as shown by the SPM analysis (p<0.001, *Figure 4*) but also when averaged over the entire stance phase (p<0.001, *Table 1*). $DC_{Belly}$ was markedly lower than $DC_{Tendon}$, indicating that the tendon took over the majority of the overall decoupling in both muscles (*Figure 4*). Accordingly and similarly to $DC_{Tendon}$, the SPM analysis for the overall decoupling of MTU and fascicles showed that

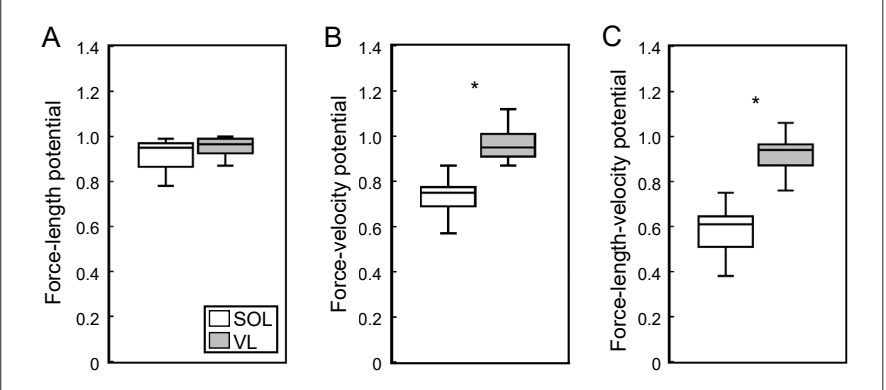

**Figure 2.** Soleus (SOL, n = 19) and vastus lateralis (VL, n = 14) force-length potential (**A**), force-velocity potential (**B**), and overall force-length-velocity potential (**C**) averaged over the stance phase of running. *Significant difference between muscles (p<0.05).

The online version of this article includes the following source data for figure 2:

**Source data 1.** Numerical data represented in the graph 2.

---

$DC_{MTU}$ of the soleus was significantly lower between 4% and 8% of the stance phase (p=0.032) and significantly higher from 20% to 57% of the stance phase and from 65% of the stance phase until lift-off compared to the vastus lateralis (p<0.001, **Figure 4**). The stance phase-averaged $DC_{MTU}$ of the soleus was significantly greater compared to the vastus lateralis as well (p<0.001, **Table 1**).

## Discussion

We mapped the operating length and velocity of the soleus and the vastus lateralis fascicles during running onto the individual force-length, force-velocity, and enthalpy efficiency-velocity curves in order to investigate physiological mechanisms for muscle force generation and muscle work production

---

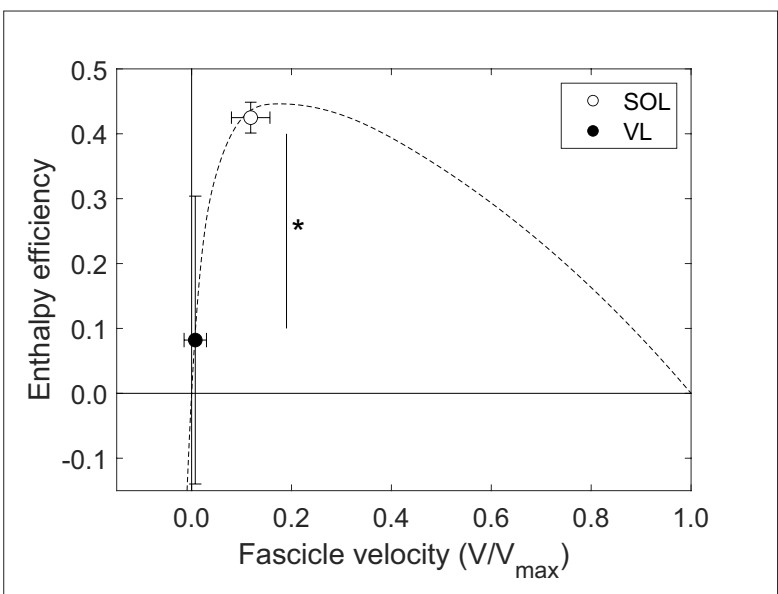

**Figure 3.** Soleus (SOL, n = 19) and vastus lateralis (VL, n = 14) enthalpy efficiency (mean ± SD) averaged over the stance phase of running onto the enthalpy efficiency-fascicle velocity relationship (dashed line). *Significant difference between muscles (p<0.05).

The online version of this article includes the following source data for figure 3:

**Source data 1.** Numerical data represented in the graph 3.

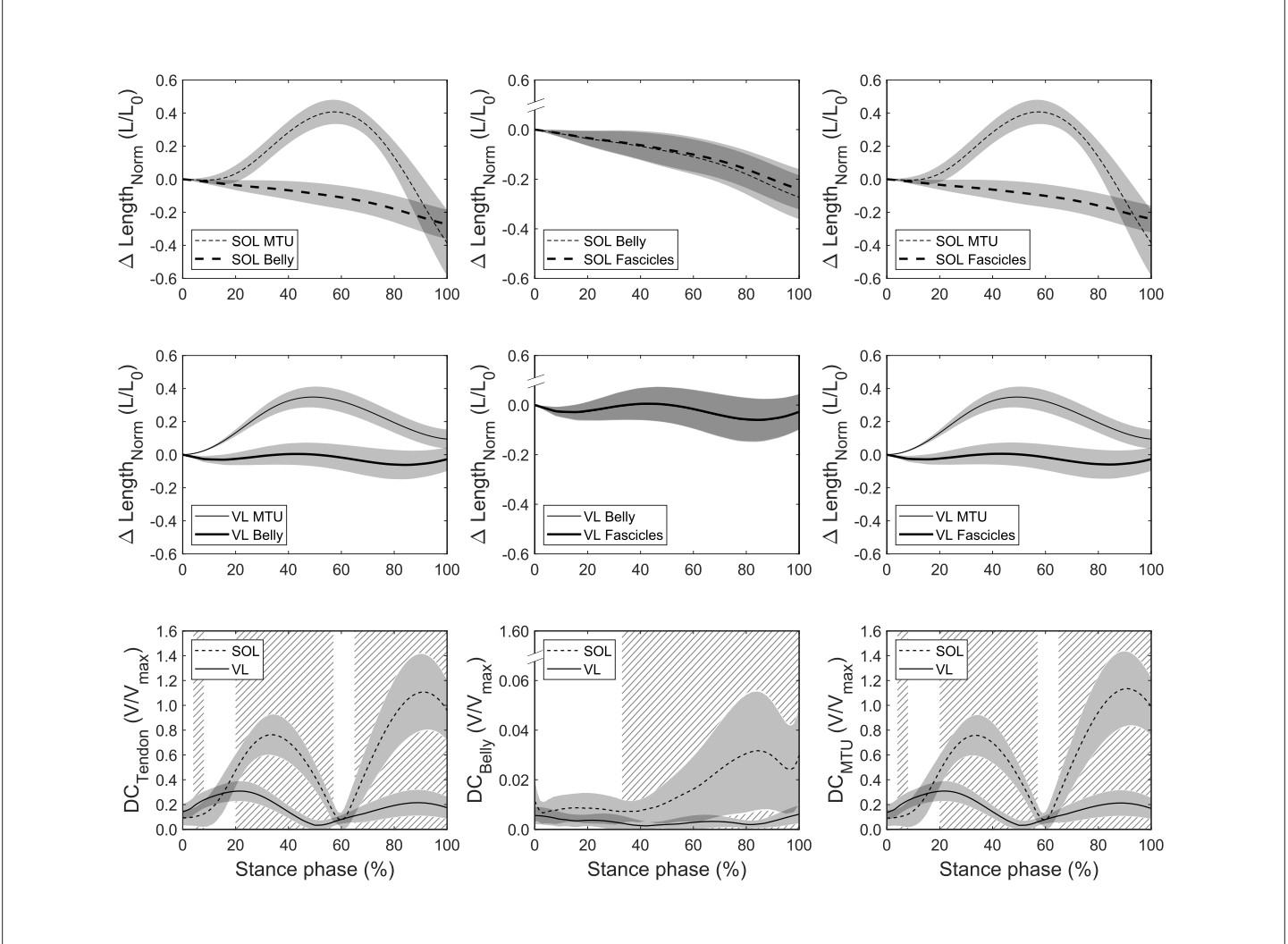

**Figure 4.** Soleus (SOL, n = 19, top row) and vastus lateralis (VL, n = 14, mid row) muscle-tendon unit (MTU) vs. belly length changes (left), belly vs. fascicle length changes (mid), and MTU vs. fascicle length changes (right) over the stance phase of running with respect to the length at touchdown (0% stance phase). Differences between curves illustrate the length-decoupling due to tendon compliance, fascicle rotation, and the overall decoupling, respectively. The bottom row shows the resulting velocity-decoupling coefficients (DCs) as the absolute velocity differences between fascicles, belly, and MTU normalized to the maximum shorting velocity (see Materials and methods). Intervals of stance with a significant difference between both muscles are illustrated as hatched areas (p<0.05).

The online version of this article includes the following source data for figure 4:

**Source data 1.** Numerical data represented in the graph 4.

**Table 1.** Average tendon ($DC_{Tendon}$), belly ($DC_{Belly}$), and muscle-tendon unit ($DC_{MTU}$) decoupling coefficients for the soleus (SOL) and vastus lateralis (VL) muscles during the stance phase of running (mean ± SD).

| | SOL (n = 19) | VL (n = 14) |
|---|---|---|
| $DC_{Tendon}$ ($V/V_{max}$) | 0.567 ± 0.128 | 0.180 ± 0.053* |
| $DC_{Belly}$ ($V/V_{max}$) | 0.016 ± 0.008 | 0.003 ± 0.002* |
| $DC_{MTU}$ ($V/V_{max}$) | 0.574 ± 0.127 | 0.179 ± 0.014* |

*Statistically significant difference between the two muscles (p<0.05).

in the two muscles. The soleus continuously shortened throughout the stance phase and produced muscular work at a shortening velocity close to the enthalpy efficiency optimum. Vastus lateralis operated with smaller length changes, almost isometrically, resulting in a high force-velocity potential, which is beneficial for economic force generation. Both muscles operated close to $L_0$, that is, at a high force-length potential. Tendon compliance was responsible for the majority of the overall decoupling of MTU and fascicles in both muscles, enabling favorable conditions for muscle force or muscle work production. Only in the soleus muscle did fascicle rotation contribute to the overall decoupling, indicating an additional, yet comparatively minor effect on the fascicle dynamics during locomotion.

The triceps surae and quadriceps muscle groups are the main contributors for locomotion and thus responsible for a great portion of the metabolic energy cost of running (*Farris and Sawicki, 2012*; *Fletcher and MacIntosh, 2015*; *Uchida et al., 2016*; *Hamner and Delp, 2013*). While the quadriceps mainly decelerates and supports body mass in the early stance phase, the triceps surae contributes to the acceleration of the center of mass during the second part of the stance phase (*Dorn et al., 2012*; *Hamner and Delp, 2013*). The soleus is the largest muscle of the triceps surae (*Albracht et al., 2008*) and the vastus lateralis of the quadriceps (*Mersmann et al., 2015*) and thus both muscles are important for the running movement. We found that the soleus actively shortened throughout the entire stance phase, indicating continuous work/energy production. The average velocity at which the soleus shortened was very close to the optimal velocity for maximal enthalpy efficiency. Enthalpy efficiency quantifies the fraction of chemical energy from ATP hydrolysis that is converted into mechanical muscular work (*Hill, 1964*; *Barclay, 2015*) with a peak at around 20% of $V_{max}$ (*Hill, 1939*; *Barclay et al., 1993*). Consequently, the mechanical work performed by the soleus muscle, being essential during running (*Arampatzis et al., 1999*; *Stefanyshyn and Nigg, 1998*; *Hamner and Delp, 2013*; *Lai et al., 2015*) and high enough in magnitude to significantly influence the overall metabolic energy cost of locomotion (*Bohm et al., 2019*; *Sawicki et al., 2020*; *Beck et al., 2019*), was generated at a high enthalpy efficiency (94% of maximum efficiency). Considering that also the soleus force-length potential was close to the maximum (0.92) and that a high potential may decrease the active muscle volume for a given muscle force (*Beck et al., 2019*; *Biewener and Roberts, 2000*; *Fletcher and MacIntosh, 2017*), our results provide evidence of a cumulative contribution of two different mechanisms (high force-length potential and high enthalpy efficiency) to an advantageous muscular work production of the soleus during running. The vastus lateralis was mainly active in the first part of the stance phase and its fascicles operated with very small length changes, that is, almost isometrically, confirming earlier reports (*Bohm et al., 2018*; *Monte et al., 2020*). This indicates that the vastus lateralis dissipates and/or produces negligible amounts of mechanical energy during running, yet generating force for the deceleration and support of the body mass. The observed decoupling of the vastus lateralis MTU and fascicles showed that the deceleration of the body mass in the early stance phase was not a result of an energy dissipation by the contractile element (active stretch) but rather an energy absorption by the tendinous tissue. Tendons feature low damping characteristics, resulting in a hysteresis of only 10% (*Pollock and Shadwick, 1994*; *Bennett et al., 1986*), and, therefore, the main part of the absorbed energy of the body's deceleration is expected to be stored as elastic tendon strain energy, which is then returned later in the second part of the stance phase. The high force-length (0.93) and force-velocity (0.96) potential of the vastus lateralis muscle throughout stance indicates an energy exchange within the vastus lateralis MTU under almost optimal conditions for muscle force generation during running. Operating at high potentials reduces the active muscle volume for a given force (*Biewener and Roberts, 2000*; *Fletcher and MacIntosh, 2017*) and thus the metabolic energy cost of muscle force generation.

By actively shortening the soleus delivered energy during the entire stance phase to the skeleton, providing the main muscular work required for running. On the other side, the contractile elements of the vastus lateralis muscle did not contribute to the required muscular work and operated in concert with the elastic tendon in favor of energy storage (*Roberts and Azizi, 2011*). Our findings showed that, although the human body interacts with the ground in a spring-like manner during steady-state running to store mechanical energy (*Dickinson et al., 2000*; *Roberts and Azizi, 2011*), there are indeed muscles that operate as work generators, like the soleus, and others that promote energy conservations, like the vastus lateralis. Further, our results indicate that the fascicle operating length and velocity of the soleus muscle, the main work generator, is optimized for high enthalpy efficiency, while of the vastus lateralis muscle, which promote energy conservation, for a high potential of force

generation. The consequence of the active shortening of the soleus muscle for work production is a decrease of the force-velocity potential during the stance phase, which may increase the active muscle volume and shortening-related cost (*Hill, 1938*; *Fenn, 1924*; *Smith et al., 2005*; *He et al., 2000*). However, the soleus muscle features shorter fascicles ($L_0$ = 41 mm) compared to the vastus lateralis muscle ($L_0$ = 94 mm), and, for this reason, a given force generated by the soleus is energetically less expensive (*Biewener and Roberts, 2000*). The specific morphology of the soleus muscle certainly compensates for the reductions of the force-velocity potential and provides advantages for its function as work generator during submaximal steady-state running. Furthermore, operating around the 'sweet spot' of the shortening velocity for high enthalpy efficiency facilitates the economical muscular work production, while either a too high or a too low shortening velocity would be disadvantageous. The advantageous operating conditions specific for soleus and vastus lateralis during submaximal running shown here for a moderate speed of 2.5 m/s seem to persist at faster running speeds as well. This is indicated by recent evidence of a comparable muscle operating length and velocity of the soleus (*Lai et al., 2015*) and vastus lateralis (*Monte et al., 2020*) over a broad range of running speeds, respectively. In addition, the operating behavior of both muscles seems to reflect their respective muscle group. The gastrocnemius muscles, as the second largest plantar flexors, have been shown to operate at a length similar to soleus and only at slightly higher velocities (*Lai et al., 2018*), suggesting efficient work production too. For the other monoarticular knee extensors, vastus medialis and intermedius, the resting fascicle length is about similar to the vastus lateralis (*Ward et al., 2009*), and, since they share the same single patellar tendon, we also do not expect that those muscles operate substantially different (*Arnold et al., 2013*), that is, likewise at a high force potential.

The almost optimal conditions for muscular work production and muscle force generation of the soleus and vastus lateralis were a result of an effective decoupling between MTU and fascicle length that was regulated by an appropriate muscle activation. For the soleus, the activation level increased in the first part of stance phase, contracting the muscle while the MTU increased in length. This activation pattern not only prevented the muscle to be stretched but also induced continuous shortening around the plateau of the force-length curve at a high enthalpy efficiency. The respective high $DC_{Tendon}$ further indicates that a part of the body's mechanical energy was stored as strain energy in the Achilles tendon in addition to the generated work by fascicle shortening. During MTU shortening (propulsion phase), the soleus EMG activity decreased and the tendon recoiled, enabling the high shortening velocities of the MTU while maintaining the fascicle operating conditions close to the efficiency optimum. The simultaneous release of the stored strain energy from the tendon further added to the ongoing muscle work production, that is, energy amplification. The vastus lateralis muscle showed higher levels of activation during the initial part of the stance phase and earlier deactivation than soleus. The timing and level of activation regulated the decoupling within the vastus lateralis MTU during the body mass deceleration in a magnitude that the lengthening and shorting of the MTU was fully accomplished by the tendinous tissue. Consequently, the vastus lateralis fascicles operated at a high force-length-velocity potential and the body's energy was stored within the MTU. Although being substantial for soleus and vastus lateralis, the SPM analysis revealed higher values of $DC_{Tendon}$ for soleus during the major part of the stance phase (average value for soleus 0.57 $V/V_{max}$ and vastus lateralis 0.18 $V/V_{max}$), indicating a greater decoupling within the soleus MTU compared to the vastus lateralis MTU. In the soleus muscle, fascicle rotation (changes in pennation angle) had an additional effect on the overall decoupling between MTU and fascicles. The results showed an increase in $DC_{Belly}$ in the second part of the stance phase where the soleus belly velocity was high during the MTU shortening. However, the decoupling by the fascicle rotation was considerably smaller compared to the tendon decoupling. Over the stance phase, belly and tendon decoupling were 1.6% $V_{max}$ and 57% $V_{max}$ and during the MTU shortening phase 2.6% $V_{max}$ and 72% $V_{max}$, respectively, suggesting a rather minor functional role of fascicle rotation during submaximal running. In the vastus lateralis, fascicle rotation was virtually absent and consequently $DC_{Belly}$ values showed no relevant decoupling effect at all.

Note that because of the extensive experimental protocol for each muscle it was not possible to measure soleus and vastus lateralis in the same participants. However, both groups are a representative sample and no significant differences were found in anthropometrics and relevant gait parameters. Furthermore, for the determination of the vastus lateralis force-length curve, the muscle was not isolated, hence the curve also includes the contribution of the vastus medialis and intermedius. The underlying assumption for this approach is that the force-length curves of these three synergistic

knee extensors are comparable, which is supported by the study of Herzog et al. (*Herzog et al., 1990*). Besides, it is currently not possible to measure enthalpy efficiency directly during running. Instead, we used an experimentally determined efficiency-velocity curve reported by Hill (*Hill, 1964*) and confirmed by others (*Barclay et al., 2010*) to relate the measured operating fascicle velocities of both muscles to the enthalpy efficiency. We were also not able to directly measure $V_{max}$ of both muscles and despite using a biologically funded value, its choice affects the force-velocity potential and enthalpy efficiency. However, when conducting a sensitivity analysis by substantially reducing or increasing $V_{max}$ by 30%, the force-velocity potential of vastus lateralis only changed for $V_{max}$ -30% and $V_{max}$ +30 % from 0.96 to 0.94 and 0.98 and of soleus from 0.63 to 0.52 and 0.65 while the enthalpy efficiency changed from 0.082 to 0.081 and 0.016 for vastus lateralis and from 0.425 to 0.439 and 0.403 for soleus, respectively, without an impact on the significance of the differences between muscles (potential $p<0.001$, efficiency $p<0.001$). These results support the robustness of our primary outcomes and strengthen our conclusions.

In conclusion, the present study demonstrated that during the stance phase of steady-state running, when the human body interacts with the environment in a spring-like manner, the soleus muscle acts as work generator and the vastus lateralis muscle as energy conservator. Furthermore, our findings provide evidence that the soleus operates under conditions optimal for muscular work production (i.e., high force-length potential and high enthalpy efficiency) and the vastus lateralis under conditions optimal for muscle force generation (i.e., high force-length and high force-velocity potential).

## Materials and methods
### Participants and experimental design
Thirty-three physically active adults, accustomed to regular running on a recreational basis (i.e., no competitive runners), were included in the present investigation. None of the participants reported any history of neuromuscular or skeletal impairments in the 6 months prior to the recordings. The ethics committee of the university approved the study (EA2/076/15), and the participants gave written informed consent in accordance with the Declaration of Helsinki. From the right leg, either the soleus (n = 19, 29 ± 6 years, 177 ± 9 cm, 69 ± 9 kg, seven females) or vastus lateralis (n = 14, age 28 ± 4 years, height 179 ± 7 cm, body mass 75 ± 8 kg, three females) muscle fascicle length, fascicle pennation angle, and EMG activity were recorded during running on a treadmill at 2.5 m/s. Corresponding MTU lengths were calculated from the kinematic data and individually measured tendon lever arms. We further assessed the soleus and vastus lateralis force-fascicle length and force-fascicle velocity relationship to calculate the force-length and force-velocity potential of the soleus and the vastus lateralis muscle fascicles during running. The operating fascicle velocity was additionally mapped on the enthalpy efficiency-velocity relationship to assess the enthalpy efficiency of both muscles. The contribution of the decoupling of the fascicle length and velocity from the MTU to the operating force potential and enthalpy efficiency at the level of tendon and muscle belly during running was examined for both muscles as well. All data for one participant were collected on the same day and sensors (EMG, ultrasound, reflective markers) remained attached between the different parts of the experiment.

### Joint kinematics, fascicle behavior, and electromyographic activity during running
After a familiarization phase, a 4 min running trial on a treadmill (soleus: h/p cosmos mercury, Isny, Germany; vastus lateralis: Daum electronic, ergo_run premium8, Fürth, Germany) was performed and kinematics of the right leg were captured by a Vicon motion capture system (version 1.8, Vicon Motion Systems, Oxford, UK, 250 Hz) using an anatomically referenced reflective marker setup (greater trochanter, lateral femoral epicondyle and malleolus, fifth metatarsal, and tuber calcanei). The kinematic data were used to determine the touchdown of the foot and the toe-off as consecutive minima in knee joint angle over time (*Fellin et al., 2010*). Furthermore, the kinematics of the ankle and knee joint served to calculate the MTU length change of the soleus and vastus lateralis during running as the product of ankle joint angle changes and Achilles tendon lever arm as well as knee joint angle changes and patellar tendon lever arm (*Lutz and Rome, 1996*), respectively. We used the ultrasound-based tendon-excursion method for the Achilles tendon lever arm determination (*An*

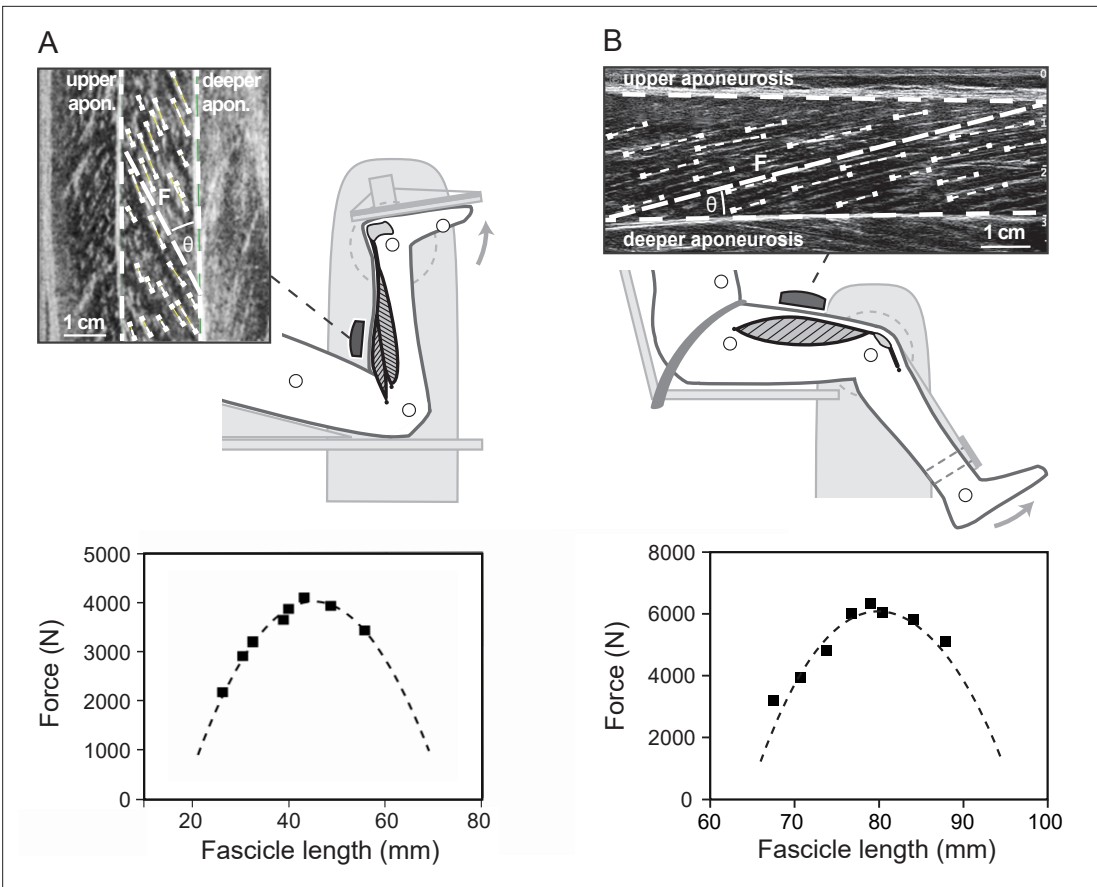

**Figure 5.** Experimental setup for the determination of the soleus (**A**) and vastus lateralis (**B**) force-fascicle length relationship. Maximum isometric plantar flexions (MVC) at eight different joint angles were performed on a dynamometer. During the MVCs, ultrasound images of the soleus and vastus lateralis were recorded and a representative muscle fascicle length (F) was calculated based on multiple fascicle portions (short dashed lines). Accordingly, an individual force-fascicle length relationship for the soleus and vastus lateralis muscle was derived from the MVCs (squares) by means of a second-order polynomial fit (dashed line, bottom graphs, MVCs and curves of one representative participant).

*et al., 1984*). The patellar tendon lever arm was measured using magnetic resonance imaging in fully extended knee joint position and calculated as a function of the knee joint angle change using the data by Herzog and Read (*Herzog and Read, 1993*; for a detailed description of both tendon lever arm measurements, see *Bohm et al., 2019*; *Bohm et al., 2018*; *Bohm et al., 2021*). The initial soleus and vastus lateralis MTU length was calculated based on the regression equation provided by Hawkins and Hull (*Hawkins and Hull, 1990*) at neutral ankle joint angle for the soleus MTU and at touchdown for the vastus lateralis MTU. During the running trial, ultrasound images of either the soleus or vastus lateralis muscle fascicles were recorded synchronously to the kinematic data (soleus: Aloka Prosound Alpha 7, Hitachi, Tokyo, Japan, 6 cm linear array probe, UST-5713T, 13.3 MHz, 146 Hz; vastus lateralis: My Lab60, Esaote, Genova, Italy, 10 cm linear array probe LA923, 10 MHz, 43 Hz). The ultrasound probe was mounted over the medial aspect of the soleus muscle belly or on the vastus lateralis muscle belly (≈50% of femur length) using a custom anti-skid neoprene-plastic cast. The fascicle length was post-processed from the ultrasound images using a self-developed semi-automatic tracking algorithm (*Marzilger et al., 2018*) that calculated a representative reference fascicle on the basis of multiple muscle fascicle portions identified from the entire displayed muscle (for details, see *Bohm et al., 2018*; *Marzilger et al., 2018*; *Figure 5*). Visual inspection of each image was conducted and corrections were made if necessary. At least nine steps were analyzed for each participant and then averaged (*Bohm et al., 2018*; *Giannakou et al., 2011*). The pennation angle was calculated as the angle between the deeper aponeurosis and the reference fascicle (*Figure 5*). The length changes of the muscle belly of soleus and vastus lateralis were calculated as the differences of consecutive products of fascicle length and the respective cosine of the pennation angle (*Fukunaga et al., 2001*). Note

that this does not give the length of the entire soleus or vastus lateralis muscle belly but rather the projection of the instant fascicle length onto the plane of the MTU, which can be used to calculate the changes of the belly length (*Bohm et al., 2019*). The velocities of fascicles, belly, and MTU were calculated as the first derivative of the lengths over time.

Surface EMG of the vastus lateralis and the soleus were measured by means of a wireless EMG system (Myon m320RX, Myon AG, Baar, Switzerland, 1000 Hz). A fourth-order high-pass Butterworth filter with 50 Hz cut-off frequency, a full-wave rectification, and then a low-pass filter with 20 Hz cut-off frequency were applied to the raw EMG data. The EMG activity was averaged over the same steps that were analyzed for the soleus parameters and for the vastus lateralis over 10 running steps. EMG values were then normalized for each participant to the maximum obtained during a individual maximum voluntary contraction.

## Assessment of the force-length, force-velocity, and enthalpy efficiency-velocity relationship

To determine the soleus and the vastus lateralis force-length relationship, eight maximum voluntary plantar flexion or knee extension contractions (MVCs) in different joint angles were performed with the right leg on an isokinetic dynamometer (Biodex Medical, Syst. 3, Inc, Shirley, NY), following a standardized warm-up (*Bohm et al., 2019*; *Bohm et al., 2018*; *Nikolaidou et al., 2017*; *Figure 5*). For the plantar flexion MVCs, the participants were placed in prone position with the knee in fixed flexed position (~120°) to restrict the contribution of the bi-articular m. gastrocnemius to the plantar flexion moment (*Hof and van den Berg, 1977*) and the joint angles were set in a randomized equally distributed order ranging from 10° plantar flexion to the individual maximum dorsiflexion angle. Regarding the knee extensions, participants were seated with a hip joint angle of 85° to reduce the contribution of the bi-articular m. rectus femoris (*Herzog et al., 1990*), while the knee joint angle ranged between 20° to 90° knee joint angle (0° = knee extended) in randomly ordered 10° intervals. The resultant moments at the ankle and knee joint were calculated under consideration of the effects of gravitational and passive moments and any misalignment between joint axis and dynamometer axis using an established inverse dynamics approach (*Arampatzis et al., 2005*; *Arampatzis et al., 2004*). The required kinematic data were recorded during the MVCs based on anatomically referenced reflective markers (medial and lateral malleoli and epicondyle, calcaneal tuberosity, second metatarsal, and greater trochanter) by a Vicon motion capture system (250 Hz). Furthermore, the contribution of the antagonistic moment produced by tibialis anterior during the plantar flexion MVCs or by the hamstring muscles during the knee extension MVCs was taken into account by means of an EMG-based method according to Mademli et al. (*Mademli et al., 2004*), considering the force-length dependency of the antagonists (*Bohm et al., 2021*). The force applied to the Achilles or patellar tendon during the plantar flexion or knee extension MVCs was calculated as quotient of the joint moment and individual tendon lever arm, respectively. The soleus or the vastus lateralis fascicle behavior during the MVCs was synchronously captured by ultrasonography and fascicle length was determined using the same methodology described above (*Figure 5*). Accordingly, an individual force-fascicle length relationship was calculated for soleus or vastus lateralis by means of a second-order polynomial fit and $F_{max}$ and $L_0$ was derived, respectively (*Figure 5*).

The force-velocity relationship of the soleus and the vastus lateralis muscle was further assessed using the classical Hill equation (*Hill, 1938*) and the muscle-specific $V_{max}$ and constants of $a_{rel}$ and $b_{rel}$. For $V_{max}$, we took values of human soleus and vastus lateralis type 1 and 2 fibers measured in vitro at 15 °C reported by Luden et al. (*Luden et al., 2008*). The values were then adjusted (*Ranatunga, 1984*) for physiological temperature conditions (37 °C) and an average fiber type distribution of the human soleus (type 1 fibers: 81%, type 2: 19%) and vastus lateralis muscle (type 1 fibers: 37%, type 2: 63%) reported in literature (*Johnson et al., 1973*; *Luden et al., 2008*; *Edgerton et al., 1975*; *Larsson and Moss, 1993*) was the basis to derive a representative value of $V_{max}$. For the soleus muscle under the in vivo condition, $V_{max}$ was calculated as 6.77 $L_0$/s and for the vastus lateralis as 11.51 $L_0$/s. For $L_0$, we then referred to the individually measured optimal fascicle length (described above, *Figure 5*). The constant $a_{rel}$ was calculated as 0.1 + 0.4 FT, where FT is the fast twitch fiber type percentage, which then equals to 0.175 for the soleus and 0.351 for the vastus lateralis (*Winters and Stark, 1985*; *Winters and Stark, 1988*). The product of $a_{rel}$ and $V_{max}$ gives the constant $b_{rel}$ as 1.182 for the soleus and 4.042 for the vastus lateralis (*Umberger et al., 2003*). Based on the assessed force-length and

force-velocity relationships, we calculated the individual force-length and force-velocity potential of both muscles as a function of the fascicle operating length and velocity during the stance phase of running. The product of both potentials then gives the overall force-length-velocity potential.

Furthermore, we determined the enthalpy efficiency-velocity relationship for the soleus and the vastus lateralis muscle fascicles in order to calculate the enthalpy efficiency of both muscles as a function of the fascicle operating velocity during running. For this purpose, we used the experimental efficiency values provided by the paper of *Hill, 1964* in *Table 1* for $a/P_0 = 0.25$ (*Hill, 1964*). By means of the classical Hill equation (*Hill, 1938*), we then transposed the original efficiency values that were presented as a function of relative load (relative to maximum tension) to shortening velocity (normalized to $V_{max}$). The values of enthalpy efficiency and shortening velocity were then fitted using a cubic spline, giving the right-skewed parabolic-shaped curve with a peak efficiency of 0.45 at a velocity of 0.18 $V/V_{max}$. The resulting function was then used to calculate the enthalpy efficiency of the soleus and the vastus lateralis during running based on the average value of the fascicle velocity over stance, accordingly.

## Assessment of decoupling within the MTU

To quantify the decoupling of fascicle, belly, and MTU velocities over the time course of stance, we calculated a decoupling coefficient to account for the tendon compliance ($DC_{Tendon}$, equation 1), fascicle rotation ($DC_{Belly}$, equation 2), as well as for the overall decoupling of MTU and fascicle velocities that includes both components ($DC_{MTU}$, equation 3).

$$DC_{Tendon}\left(t\right) = \left|V_{MTU}\left(t\right) - V_{Belly}\left(t\right)\right|/V_{max} \tag{1}$$

$$DC_{Belly}\left(t\right) = |V_{Belly}\left(t\right) - V_{Fascicle}\left(t\right)|/V_{max} \tag{2}$$

$$DC_{MTU}\left(t\right) = |V_{MTU}\left(t\right) - V_{Fascicle}\left(t\right)|/V_{max} \tag{3}$$

where *V(t)* is the velocity at each percentage of the stance phase (i.e. t = 0, 1, …, 100% stance). We introduced these new decoupling coefficients because previously suggested decoupling ratios (i.e., tendon gearing = $V_{MTU}/V_{Belly}$, belly gearing [or architectural gear ratio] = $V_{Belly}/V_{Fascicle}$, MTU gearing = $V_{MTU}/V_{Fascicle}$; *Azizi et al., 2008*; *Wakeling et al., 2011*) may feature limitations for the application under in vivo conditions, that is, considering that muscle belly and fascicle velocities may be very close to or even zero during functional tasks as walking and running (*Bohm et al., 2019*; *Bohm et al., 2018*), which results in non-physiological gear ratios.

## Statistics

A t-test for independent samples was used to test for group differences in anthropometric characteristics, temporal gait parameters, and differences between the soleus and the vastus lateralis fascicle belly, MTU, and EMG parameters. The Mann–Whitney U test was applied in case the assumption of normal distribution, tested by the Kolmogorov–Smirnov test with Lilliefors correction, was not met. The level of significance was set to $\alpha = 0.05$, and the statistical analyses were performed using SPSS (IBM Corp., version 22, NY). Furthermore, SPM (independent samples t-test, $\alpha = 0.05$) was used to test for differences between the $DC_{Tendon}$, $DC_{Belly}$, and $DC_{MTU}$ of the soleus and the vastus lateralis throughout the stance phase of running. SPM was conducted using the software package spm1D (version 0.4, http://www.spm1d.org; *Pataky, 2012*).

## Acknowledgements

Funding for this research was supplied by the German Federal Institute of Sport Science (grant no. ZMVI14-070604/17-18). The magnetic resonance image acquisition was funded by the foundation Stiftung Oskar-Helene-Heim. We further acknowledge support by the German Research Foundation (DFG) and the Open Access Publication Fund of Humboldt-Universität zu Berlin.

# Additional information

## Funding

| Funder | Grant reference number | Author |
|--------|------------------------|--------|
| German Federal Institute of Sport Science | ZMVI14-070604/17-18 | Adamantios Arampatzis |
| Stiftung Oskar Helene Heim | | Sebastian Bohm |

The funders had no role in study design, data collection and interpretation, or the decision to submit the work for publication.

## Author contributions

Sebastian Bohm, Conceptualization, Data curation, Formal analysis, Investigation, Methodology, Project administration, Visualization, Writing - original draft, Writing – review and editing; Falk Mersmann, Data curation, Methodology, Writing – review and editing; Alessandro Santuz, Methodology, Writing – review and editing; Arno Schroll, Methodology, Software, Writing – review and editing; Adamantios Arampatzis, Conceptualization, Funding acquisition, Methodology, Supervision, Writing - original draft, Writing – review and editing

## Author ORCIDs

Sebastian Bohm ORCID http://orcid.org/0000-0002-5720-3672
Falk Mersmann ORCID http://orcid.org/0000-0001-7180-7109
Alessandro Santuz ORCID http://orcid.org/0000-0002-6577-5101
Adamantios Arampatzis ORCID http://orcid.org/0000-0002-4985-0335

## Ethics

Human subjects: The ethics committee of the Humboldt-Universität zu Berlin approved the study and the participants gave written informed consent in accordance with the Declaration of Helsinki.

## Decision letter and Author response

Decision letter https://doi.org/10.7554/eLife.67182.sa1
Author response https://doi.org/10.7554/eLife.67182.sa2

# Additional files

## Supplementary files

• Transparent reporting form

## Data availability

The final processed data can be found at: https://doi.org/10.6084/m9.figshare.14046749.

The following dataset was generated:

| Author(s) | Year | Dataset title | Dataset URL | Database and Identifier |
|-----------|------|---------------|-------------|--------------------------|
| Bohm S, Mersmann F, Santuz A, Schroll A, Arampatzis A | 2021 | Data_Muscle-specific economy of force generation and efficiency of work production during human running | https://doi.org/10.6084/m9.figshare.14046749 | figshare, 10.6084/m9.figshare.14046749 |

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
