## [Decision Letter]

**Acceptance summary:**

This manuscript provides novel insights into the role and interactions of the vastus lateralis and soleus in running and the complementary roles they play.

**Decision letter after peer review:**

Thank you for submitting your article "Muscle-specific economy of force generation and efficiency of work production during human running" for consideration by *eLife*. Your article has been reviewed by 2 peer reviewers, and the evaluation has been overseen by a Reviewing Editor and Carlos Isales as the Senior Editor. The reviewers have opted to remain anonymous.

Essential revisions:

1) The data were measured during treadmill running at 2.5 m/s, which is 9km/h (a time of 6 min 40s /km). This is a rather slow running speed and presumably all participants were able to run at that speed comfortably. However, the function of the muscles under examination might also depend on the running speed as energy costs depend on running speed as well. With increasing running speed ground contact times (stance times) might decrease affecting the contraction velocity of soleus muscle fascicles. In consequence the results of the present study might reflect specifically the role of soleus and vastus lateralis during running on a treadmill at 2.5 m/s but an extrapolation to other conditions remains vague as muscle function may depend on running velocity , as well as on other factors like e.g. compliance of ground which is considerably different when running on a treadmill compared to e.g. hard soil. The authors should have discussed this issue in more detail along with data from other studies that already showed MTU behavior and fascicle length over different running speeds.

2) As the authors aim to elucidate the role of two muscles (soleus and vastus lateralis) in one well defined movement (running on a treadmill at 2.5 m/s) it surprises that they have designed their experiment with two different groups. The authors show that groups are comparable, however they should provide a conclusive rationale for this specific characteristic of their experimental design as apparently it increases variability and hamper comparability between results (e.g. stance and swing times between groups).

3) Throughout the manuscript you use "activation" and "activity" almost synonymously. Please consider to use activation when you want to express that the muscle is activated by the CNS and activity to express that the muscle is active (contract). Thus, in the Introduction , L.39 it should consequently read "activity" instead of "activation ,.…

4) Please add the athletic background to your description of your participants. It would be useful to know especially how many have a running athletic background as this can influence running technique.

5) I consider the sophisticated modeling one of the major strengths of your study , however at the same time it can be a weakness as the error in the calculations of your primary outcome variables becomes difficult to predict. I suggest to address this issue within the discussion.

6) Distilling the entire knee and ankle extensor group actions to 1 fascicle in the vastus lateralis and soleus is quite an assumption. I understand that they are the largest muscles in each group, but is it certain that they provide more biomechanical actions and/or metabolic cost about their joints than other muscles (ex. sol vs gastroc)? The respective muscles compromise less than half of the muscle mass per leg joint extensors (Biewener et al. 2004), neighboring muscles cross a different number of joints (uni- and bi-articular) indicating that their biomechanical actions are different to some extent.

7) Because muscles, like the soleus, have multiple compartments with complicated architecture, tracking only 1 fascicle may not represent fascicles mechanics in other compartments of the soleus.

8) Are the metabolic costs of performing work separate from the metabolic costs of producing work and accounting for force-length-velocity potential? Consider actin-myosin crossbridge ATP turnover, if a crossbridge cycles and shortens the sarcomere, does it use more ATP to produce work, to operate at a shorter length-velocity, and to perform work? Making implications regarding force-length-velocity potential and enthalpy efficiency confuses me and makes me think that there are separate costs that one could add up to determine the ATP use in a muscle contraction.

9) I was surprised that his study did not take any metabolic cost measures. The study's hypotheses both discuss the metabolic cost of each muscle group, but the authors did not study the metabolic cost of the muscles, only the mechanics of 1 fascicle per muscle.

10) Some of the comparisons between Sol and VL do not seem valuable. Do the authors need to compare soleus to vastus lateralis mechanics to make a statement about each muscle? Could the authors justify a threshold, such as 0.85 force-velocity potential that differentiates isometric versus concentric actions and describe their results according to the muscle's action?

11) Discussion lines 181-184: the authors never measured VL force-length, SOL/VL force-velocity, or SOL/VL efficiency values – rather the authors built their study on many layers of unmeasurable assumptions that severely limit the use of this study

---

## [Author Response]

Essential revisions:1) The data were measured during treadmill running at 2.5 m/s, which is 9km/h (a time of 6min40s /km). This is a rather slow running speed and presumably all participants were able to run at that speed comfortably. However, the function of the muscles under examination might also depend on the running speed as energy costs depend on running speed as well. With increasing running speed ground contact times (stance times) might decrease affecting the contraction velocity of soleus muscle fascicles. In consequence the results of the present study might reflect specifically the role of soleus and vastus lateralis during running on a treadmill at 2.5 m/s but an extrapolation to other conditions remains vague as muscle function may depend on running velocity , as well as on other factors like e.g. compliance of ground which is considerably different when running on a treadmill compared to e.g. hard soil. The authors should have discussed this issue in more detail along with data from other studies that already showed MTU behavior and fascicle length over different running speeds.

Thank you for this comment. For the soleus and VL, evidence suggests that their fascicle behaviors are almost unaffected by running speed. The study by Monte et al. [1] showed that during higher speeds of 2.8, 3.6 and 4.4 m/s, the shortening velocity of the VL remained very close to zero, i.e. 3.1, 1.8 and −2.8 %V/V_max_, respectively. Similarly, the operating length remained close to optimal length at the higher speeds, i.e. 1.03, 1.06 and 1.10 L/L_0_. For the soleus, Lai et al. [2] reported for a very broad range of speeds (2.0-5.0 m/s) that, although shortening velocity increased significantly, the magnitude of increase was rather low (2.0 m/s: 1.59 lsms vs. 5.0 m/s: 2.60 lsms; lsm resting muscle fascicle length measured during static standing). Given a muscle shorting velocity of 1.7 lsms at a speed of 2.5 m/s, extracted from the data of Lai et al., the increase to 5.0 m/s (2.60 lsms) equals a factor of 1.53. Applying this factor to our soleus muscle shortening velocity at 2.5 m/s, the predicted enthalpy efficiency at a speed of 5.0 m/s would be 0.446. This would still cover the plateau on the efficiency-velocity curve (see Author response image 1) and would be comparable to the 2.5 m/s observed in our study (2.3% higher). Regarding the soleus fascicle length, Lai et al. reported a decreased with increasing speed, but the authors themselves argued that the fascicles would remain on the plateau of the force length curve (2.0 m/s: 1.04 lsm and 5.0 m/s: 0.92 lsm). These findings demonstrate that in a range of slow to fast running, the soleus and VL muscles do not substantially change their operating profiles.

**Author response image 1. sa2fig1:** Predicted efficiency for running at 5.0 m/s.

Regarding the effect of external factors suggested by the reviewers (treadmill and mechanical ground properties), the study by Cronin et al. [3] showed that neither shod vs. barefoot nor treadmill vs. overground walking and running significantly affected the soleus fascicle operating range and velocity. In further support, the study by Hollville et al. [4] indicated that between different surfaces with different mechanical properties, the gastrocnemius medialis and VL fascicle behavior during jumping was unaffected by the type of surface. Therefore, we are confident that our results can be generalized to running on common running surfaces.We added the following paragraph to the limitations section (page: 6, line: 234):

“The advantageous operating conditions specific for soleus and vastus lateralis during submaximal running shown here for a moderate speed of 2.5 m/s seem to persist at faster running speeds as well. This is indicated by recent evidence of a comparable muscle operating length and velocity of the soleus [5] and vastus lateralis [6] over a broad range of running speeds, respectively.”

2) As the authors aim to elucidate the role of two muscles (soleus and vastus lateralis) in one well defined movement (running on a treadmill at 2.5 m/s) it surprises that they have designed their experiment with two different groups. The authors show that groups are comparable, however they should provide a conclusive rationale for this specific characteristic of their experimental design as apparently it increases variability and hamper comparability between results (e.g. stance and swing times between groups).

We agree with the reviewers’ comment that this can be seen as a potential limitation of the study. The reason not to measure both muscles in the same participants was because of the extensive experimental protocol for each muscle, including separate assessment stations: i. assessment of force-length relationship on the dynamometer (kinematics, EMG, ultrasound, dynamometry), ii. assessment of tendon lever arm (for VL using MRI in an external hospital lab), iii. assessment during running (kinematics, EMG, ultrasound). Due to the high overall effort for both measurements (VL and soleus) most participants would not have been willing to participate. ­­­­However, the participants in each group were chosen randomly and no significant differences were found in anthropometrics and relevant gait parameters as reported.

We now added this aspect to the limitation section of the revised manuscript (page: 7, line: 273):

“Note that because of the extensive experimental protocol for each muscle, it was not possible to measure both soleus and VL in the same participants. However, both groups are a representative sample and no significant differences between groups were found in anthropometrics and relevant gait parameters.”

3) Throughout the manuscript you use "activation" and "activity" almost synonymously. Please consider to use activation when you want to express that the muscle is activated by the CNS and activity to express that the muscle is active (contract). Thus, in the Introduction , L.39 it should consequently read "activity" instead of "activation ,.…

Thanks for your comment. The wording was changed throughout the manuscript.

4) Please add the athletic background to your description of your participants. It would be useful to know especially how many have a running athletic background as this can influence running technique.

The participants in both groups (VL and soleus group) were regularly physically active and accustomed to regular running on a recreational basis (i.e. no competitive runners). This information was added to the revised manuscript (page: 8, line: 302).

5) I consider the sophisticated modeling one of the major strengths of your study , however at the same time it can be a weakness as the error in the calculations of your primary outcome variables becomes difficult to predict. I suggest to address this issue within the discussion.

Thank you for this important comment. To account for the reviewers’ comment, we could argue that the choice of V_max_ might be most crucial to the primary outcome variables, because it may affect the force-velocity potential and enthalpy efficiency of both muscles. In our view, it is barely possible to obtain valid measurements of V_max_ in vivo in humans. Thus, we defined V_max_ based on reports of in vitro fiber testing of the human soleus and VL, which we then corrected for temperature effects and fiber type distribution in the whole muscle, resulting in a biologically founded value of V_max_. However, when conducting a sensitivity analysis by substantially reducing or increasing V_max_ by 30%, the force-velocity potential of VL only changed for V_max_-30% and V_max_+30% from 0.96 to 0.94 and 0.98 and of soleus from 0.63 to 0.52 and 0.65 while the enthalpy efficiency changed from 0.082 to 0.081 and 0.016 for VL and from 0.425 to 0.439 and 0.403 for soleus, respectively, without an impact on the significance of the differences between muscles (potential p < 0.001, efficiency p < 0.001). These results support the robustness of our primary outcomes and strengthen our conclusions.

The following paragraph was added as a part of a greater limitations section in the revised manuscript (page: 8, line: 282):

“We were also not able to directly measure V_max_ of both muscles and despite using a biologically funded value, its choice affects the force-velocity potential and enthalpy efficiency. […] These results support the robustness of our primary outcomes and strengthen our conclusions.”

6) Distilling the entire knee and ankle extensor group actions to 1 fascicle in the vastus lateralis and soleus is quite an assumption. I understand that they are the largest muscles in each group, but is it certain that they provide more biomechanical actions and/or metabolic cost about their joints than other muscles (ex. sol vs gastroc)? The respective muscles compromise less than half of the muscle mass per leg joint extensors (Biewener et al. 2004), neighboring muscles cross a different number of joints (uni- and bi-articular) indicating that their biomechanical actions are different to some extent.

Thanks for this comment. A strength of our ultrasound-based fascicle length assessment is that we track many fascicle portions over the entire field of view of the probe that are then averaged to a representative fascicle length for each muscle (illustrated in Author response image 2) [7]. That means that we did not track only one fascicle but considered multiple fascicles from the muscle belly to characterize the respective muscles fascicle length.

**Author response image 2. sa2fig2:** Ultrasound-based determination of muscle fascicle length. upper (uA) and deeper (dA) aponeurosis, representative reference fascicle (rF) calculated from all identified visible features of multiple fascicles (F).

Regarding the second part of the reviewers’ comment, we agree that more muscles are involved and contribute to the metabolic cost. Still, for the plantar flexors it is obvious that the soleus is substantially larger and features a greater PCSA compared to the gastrocnemii [8,9]. Furthermore, although the gastrocnemius muscles span also the knee joint, their fascicle length behavior during running is comparable to the continuous shortening of soleus during running [3]. Lai et al. [10] also showed no significant differences in shortening velocity of soleus and gastrocnemius lateralis, yet higher velocity of gastrocnemius medialis but only during the initial stance phase (+38%). As shown in response to comment 1, higher shortening velocities in this range would likely still cover the plateau of the efficiency-velocity curve. Thus, it can be assumed that the efficient work generation observed in soleus is comparable in the gastrocnemius muscles. The VL is the largest muscle in the quadriceps muscle group. To the best of our knowledge, there are no experimental studies on the fascicle behavior of the other muscle compartments of the quadriceps. The resting fascicle length of the monoarticular VI, VM seem to be comparable to the VL [9] and since they share the same single patellar tendon we would not expect those muscles to operate substantially different. This is supported by the modelling study of Arnold et al. that showed almost identical operating conditions of VL and VM [11]. The RF also spans the hip joint and it cannot be ruled out that this muscle operates differently, yet it is the smallest muscle in the quadriceps group [9].

We added the following sentence to the limitation section of the revised manuscript (page: 6, line: 238):

“In addition, the operating behavior of both muscles seem to reflect their respective muscle group. […] For the other monoarticular knee extensors, vastus medialis and intermedius, the resting fascicle length is about similar to the vastus lateralis [9] and since they share the same single patellar tendon we also do not expect that those muscles operate substantially different [11], i.e. likewise at a high force potential.”

7) Because muscles, like the soleus, have multiple compartments with complicated architecture, tracking only 1 fascicle may not represent fascicles mechanics in other compartments of the soleus.

According to our previous response, we used a representative fascicle length that was calculated from multiple fascicle portions and not a single fascicle (please see description and Author response image 2).

We agree with the reviewers’ comment about the complex anatomy of soleus. Using a sophisticated methodological combination of magnetic resonance imaging (MRI), diffusion tensor imaging (DTI) and 3D microdissection techniques, Bolsterlee et al. [12] recently showed that in the four soleus compartments the fascicle length were not significantly different neither when measured in short (plantar flexion) nor in long muscle length (dorsi flexion). The fascicle lengths provided by this study were also comparable in magnitude to our results. In further support, the studies of Lai and colleagues [2,10] scanned the soleus muscle during running from the lateral side (lateral-posterior compartment) while we used the medial side (medial-posterior compartment). Qualitatively, there seems no difference in the fascicle behavior of soleus throughout stance when comparing the different sides, which agrees well with the findings of fascicle length homogeneity between compartments by Bolsterlee et al. [12] mentioned before. Together these findings strengthen our view that our results represent a general soleus operating condition.

8) Are the metabolic costs of performing work separate from the metabolic costs of producing work and accounting for force-length-velocity potential? Consider actin-myosin crossbridge ATP turnover, if a crossbridge cycles and shortens the sarcomere, does it use more ATP to produce work, to operate at a shorter length-velocity, and to perform work? Making implications regarding force-length-velocity potential and enthalpy efficiency confuses me and makes me think that there are separate costs that one could add up to determine the ATP use in a muscle contraction.

Thanks for this important comment. The metabolic cost for a given muscle force depends on the muscles operating force-length-velocity potential because it determines the number of recruited muscle fibers, i.e. a high force potential reduces the active muscle volume. This means that quasi isometric contractions close the optimum of the force-length curve (e.g. vastus lateralis) are theoretically most economical for generating a given force. During steady state running, however, is not possible to operate all muscles isometrically because the human system does not perfectly conserve all the mechanical energy in each stride. Therefore, muscular work by active shortening is needed to maintain the running movement (e.g. soleus), yet it increases the metabolic cost due to a reduced force-velocity potential that will increase active muscle volume [13] as well as due to the higher cost of each fiber when actively shortening [14,15]. The metabolic cost in this case, can be minimized by generating the needed muscular work at a shortening velocity close to the maximal enthalpy efficiency and an operating length close to the optimum of the force-length curve.

We clarified the section in the introduction of revised manuscript (page: 2, line: 40):

“Running is characterized by a spring-like interaction of the body with the ground, indicating a significant conversion of the body’s kinetic and potential energy to strain energy by elongation of elastic elements, mainly tendons, that can be recovered in the propulsive second half of the stance phase [16–18]. […] The active shorting range and velocity of a muscle during movements can be reduced by its tendon and thus, an important benefit of tendon elasticity is a reduction in the metabolic cost of running [19].”

9) I was surprised that his study did not take any metabolic cost measures. The study's hypotheses both discuss the metabolic cost of each muscle group, but the authors did not study the metabolic cost of the muscles, only the mechanics of 1 fascicle per muscle.

We are not aware of any method that can measure metabolic cost of individual human muscles in vivo during locomotion. Therefore, no metabolic cost measurements of neither muscle were taken. Given the underlying known causality of muscle mechanics and metabolic energetics [14,25], we decided to investigate the main muscles for human running experimentally and draw indirect conclusions on the related muscle specific energetics of running.

10) Some of the comparisons between Sol and VL do not seem valuable. Do the authors need to compare soleus to vastus lateralis mechanics to make a statement about each muscle? Could the authors justify a threshold, such as 0.85 force-velocity potential that differentiates isometric versus concentric actions and describe their results according to the muscle's action?

We agree with the reviewers that a threshold might be useful to distinguish isometric and concentric contractions. However, a justification for such a threshold seems difficult to us, i.e. any velocity beyond zero would be shortening per definition, and we are also not aware of any value that is established in the scientific community. Therefore, we would prefer not to introduce a certain threshold. Our main purpose was to compare the muscle operating conditions with respect to physiological mechanisms of force generation and work production, i.e. force-length and force-velocity potential as well as enthalpy efficiency-velocity relationship. This comparison provide evidence that the operating behavior of both muscles is in fact different.

11) Discussion lines 181-184: the authors never measured VL force-length, SOL/VL force-velocity, or SOL/VL efficiency values – rather the authors built their study on many layers of unmeasurable assumptions that severely limit the use of this study

Thank you for this comment. It is correct that for the force-length curve of the vastus lateralis, the muscle was not isolated and that the assessed force-length curve also included the contribution of the vastus medialis and intermedius (rectus femoris was supposedly eliminated because of the chosen hip angle). The underlying assumption here is that the force-length relationship of the vastus medialis and intermedius are comparable to the vastus lateralis and in fact this is strongly supported by the study of Herzog et al. (1990) [26]. We believe that this approach is more qualified compared to estimations of single muscle forces that are based on a number of questionable assumptions.

Regarding the force-velocity curves of both muscles, we already mentioned in response to comment 5 that in our view, it is barely possible to obtain valid measurements of V_max_ in vivo in humans. Thus, we assessed V_max_ by using the individual experimentally measured optimum fascicle length and based on reports of in vitro fiber testing of the human soleus and vastus lateralis, which we then corrected for temperature effects and fiber type distribution in the whole muscle, resulting in a biologically founded value of V_max_. We then used the classical Hill equation to calculate the force-velocity curves of the two muscles for the investigation the force-velocity potential during running.

We agree with the reviewers’ comment that the choice of V_max_ might by be critical because it may affect the force-velocity potential and enthalpy efficiency of both muscles. When conducting a sensitivity analysis by substantially reducing or increasing V_max_ by 30%, the force-velocity potential of VL only changed for V_max_-30% and V_max_+30% from 0.96 to 0.94 and 0.98 and of soleus from 0.63 to 0.52 and 0.65 while the enthalpy efficiency changed from 0.082 to 0.081 and 0.016 for VL and from 0.425 to 0.439 and 0.403 for soleus, respectively, without an impact on the significance of the differences between muscles (potential p < 0.001, efficiency p < 0.001). These results may support and strengthen our findings.

Currently it seems not possible to measure the enthalpy efficiency in humans in vivo during running. Accordingly, we used an efficiency-velocity curve reported by Hill (1964) [27] that was, however, experimentally determined and mapped our measured muscle operating velocities onto this curve. The characteristic efficiency curve has also been confirmed by the later study of Barclay et al. (2010) [28] for the mouse under physiological temperature conditions (see Author response image 3 and note the comparability of curves).

**Author response image 3. sa2fig3:** Comparison of efficiency-velocity curves provided in literature and assessed operating velocities of soleus (SOL) and vastus lateralis (VL).

The following paragraph was added as a part of a greater limitations section in the revised manuscript (page: 7, line: 276): “Furthermore, for the determination of the vastus lateralis force-length curve, the muscle was not isolated, hence the curve also includes the contribution of the vastus medialis and intermedius. […] These results may support the robustness of our primary outcomes and strengthen our conclusions.”

References:

1. Monte A, Baltzopoulos V, Maganaris CN, Zamparo P. 2020 Gastrocnemius Medialis and Vastus Lateralis in vivo muscle-tendon behavior during running at increasing speeds. Scand. J. Med. Sci. Sports n/a. (doi:10.1111/sms.13662)

2. Lai A, Lichtwark GA, Schache AG, Lin Y-C, Brown NAT, Pandy MG. 2015 in vivo behavior of the human soleus muscle with increasing walking and running speeds. J. Appl. Physiol. Bethesda Md 1985 118, 1266–1275. (doi:10.1152/japplphysiol.00128.2015)

3. Cronin NJ, Finni T. 2013 Treadmill versus overground and barefoot versus shod comparisons of triceps surae fascicle behaviour in human walking and running. Gait Posture 38, 528–533. (doi:10.1016/j.gaitpost.2013.01.027)

4. Hollville E, Rabita G, Guilhem G, Lecompte J, Nordez A. 2020 Effects of Surface Properties on Gastrocnemius Medialis and Vastus Lateralis Fascicle Mechanics During Maximal Countermovement Jumping. Front. Physiol. 11. (doi:10.3389/fphys.2020.00917)

5. Lai A, Lichtwark GA, Schache AG, Lin Y-C, Brown NAT, Pandy MG. 2015 in vivo behavior of the human soleus muscle with increasing walking and running speeds. J. Appl. Physiol. Bethesda Md 1985 118, 1266–1275. (doi:10.1152/japplphysiol.00128.2015)

6. Monte A, Baltzopoulos V, Maganaris CN, Zamparo P. In press. Gastrocnemius Medialis and Vastus Lateralis in vivo muscle-tendon behavior during running at increasing speeds. Scand. J. Med. Sci. Sports n/a. (doi:10.1111/sms.13662)

7. Marzilger R, Legerlotz K, Panteli C, Bohm S, Arampatzis A. 2018 Reliability of a semi-automated algorithm for the vastus lateralis muscle architecture measurement based on ultrasound images. Eur. J. Appl. Physiol. 118, 291–301. (doi:10.1007/s00421-017-3769-8)

8. Albracht K, Arampatzis A, Baltzopoulos V. 2008 Assessment of muscle volume and physiological cross-sectional area of the human triceps surae muscle in vivo. J. Biomech. 41, 2211–2218. (doi:10.1016/j.jbiomech.2008.04.020)

9. Ward SR, Eng CM, Smallwood LH, Lieber RL. 2009 Are current measurements of lower extremity muscle architecture accurate? Clin. Orthop. 467, 1074–1082. (doi:10.1007/s11999-008-0594-8)

10. Lai AKM, Lichtwark GA, Schache AG, Pandy MG. 2018 Differences in in vivo muscle fascicle and tendinous tissue behavior between the ankle plantarflexors during running. Scand. J. Med. Sci. Sports 0. (doi:10.1111/sms.13089)

11. Arnold EM, Hamner SR, Seth A, Millard M, Delp SL. 2013 How muscle fiber lengths and velocities affect muscle force generation as humans walk and run at different speeds. J. Exp. Biol. 216, 2150–2160. (doi:10.1242/jeb.075697)

12. Bolsterlee B, Finni T, D’Souza A, Eguchi J, Clarke EC, Herbert RD. 2018 Three-dimensional architecture of the whole human soleus muscle in vivo. PeerJ 6, e4610. (doi:10.7717/peerj.4610)

13. Roberts TJ, Azizi E. 2011 Flexible mechanisms: the diverse roles of biological springs in vertebrate movement. J. Exp. Biol. 214, 353–361. (doi:10.1242/jeb.038588)

14. Smith NP, Barclay CJ, Loiselle DS. 2005 The efficiency of muscle contraction. Prog. Biophys. Mol. Biol. 88, 1–58. (doi:10.1016/j.pbiomolbio.2003.11.014)

15. He Z-H, Bottinelli R, Pellegrino MA, Ferenczi MA, Reggiani C. 2000 ATP Consumption and Efficiency of Human Single Muscle Fibers with Different Myosin Isoform Composition. Biophys. J. 79, 945–961. (doi:10.1016/S0006-3495(00)76349-1)

16. Dickinson MH, Farley CT, Full RJ, Koehl M a. R, Kram R, Lehman S. 2000 How Animals Move: An Integrative View. Science 288, 100–106. (doi:10.1126/science.288.5463.100)

17. Roberts TJ, Azizi E. 2011 Flexible mechanisms: the diverse roles of biological springs in vertebrate movement. J. Exp. Biol. 214, 353–361. (doi:10.1242/jeb.038588)

18. Cavagna GA, Saibene FP, Margaria R. 1964 Mechanical work in running. J. Appl. Physiol. 19, 249–256. (doi:10.1152/jappl.1964.19.2.249)

19. Roberts TJ. 2002 The integrated function of muscles and tendons during locomotion. Comp. Biochem. Physiol. A. Mol. Integr. Physiol. 133, 1087–1099. (doi:10.1016/S1095-6433(02)00244-1)

20. Bohm S, Mersmann F, Santuz A, Arampatzis A. 2019 The force–length–velocity potential of the human soleus muscle is related to the energetic cost of running. Proc. R. Soc. B Biol. Sci. 286, 20192560. (doi:10.1098/rspb.2019.2560)

21. Bohm S, Marzilger R, Mersmann F, Santuz A, Arampatzis A. 2018 Operating length and velocity of human vastus lateralis muscle during walking and running. Sci. Rep. 8, 5066. (doi:10.1038/s41598-018-23376-5)

22. Nikolaidou ME, Marzilger R, Bohm S, Mersmann F, Arampatzis A. 2017 Operating length and velocity of human M. vastus lateralis fascicles during vertical jumping. R. Soc. Open Sci. 4, 170185. (doi:10.1098/rsos.170185)

23. Gordon AM, Huxley AF, Julian FJ. 1966 The variation in isometric tension with sarcomere length in vertebrate muscle fibres. J. Physiol. 184, 170–192.

24. Hill Archibald Vivian. 1938 The heat of shortening and the dynamic constants of muscle. Proc. R. Soc. Lond. Ser. B – Biol. Sci. 126, 136–195. (doi:10.1098/rspb.1938.0050)

25. Barclay CJ. 2015 Energetics of contraction. Compr. Physiol. 5, 961–995. (doi:10.1002/cphy.c140038)

26. Herzog W, Abrahamse SK, ter Keurs HE. 1990 Theoretical determination of force-length relations of intact human skeletal muscles using the cross-bridge model. Pflugers Arch. 416, 113–119.

27. Hill AV. 1964 The efficiency of mechanical power development during muscular shortening and its relation to load. Proc. R. Soc. Lond. B Biol. Sci. 159, 319–324. (doi:10.1098/rspb.1964.0005)

28. Barclay CJ, Woledge RC, Curtin NA. 2010 Is the efficiency of mammalian (mouse) skeletal muscle temperature dependent? J. Physiol. 588, 3819–3831. (doi:10.1113/jphysiol.2010.192799)